# A Novel Pathway Phenotype of Temporal Lobe Epilepsy and Comorbid Psychiatric Disorders: Results of Precision Nomothetic Medicine

**DOI:** 10.3390/antiox11050803

**Published:** 2022-04-20

**Authors:** Michael Maes, Décio Sabbatini Barbosa, Abbas F. Almulla, Buranee Kanchanatawan

**Affiliations:** 1Department of Psychiatry, Faculty of Medicine, Chulalongkorn University, Bangkok 10330, Thailand; drburanee@gmail.com; 2Department of Psychiatry, Medical University of Plovdiv, 4000 Plovdiv, Bulgaria; 3IMPACT—The Institute for Mental and Physical Health and Clinical Translation, School of Medicine, Deakin University, Barwon Health, Geelong 3220, Australia; 4Health Sciences Graduate Program, Health Sciences Center, State University of Londrina, Londrina 86038-440, Brazil; sabbatini2011@hotmail.com; 5Medical Laboratory Technology Department, College of Medical Technology, The Islamic University, Najaf 54001, Iraq; abbass.chem.almulla1991@gmail.com

**Keywords:** depression, schizophrenia, neuroimmune, oxidative and nitrosative stress, antioxidants, inflammation

## Abstract

No precision medicine models of temporal lobe epilepsy (TLE) and associated mental comorbidities have been developed to date. This observational study aimed to develop a precision nomothetic, data-driven comorbid TLE model with endophenotype classes and pathway phenotypes that may have prognostic and therapeutical implications. We recruited forty healthy controls and 108 TLE patients for this research and assessed TLE and psychopathology (PP) features as well as oxidative stress (OSTOX, e.g., malondialdehyde or MDA, lipid hydroperoxides, and advanced oxidation protein products) and antioxidant (paraoxonase 1 or PON1 status, -SH groups, and total radical trapping potential or TRAP) biomarkers. A large part (57.2%) of the variance in a latent vector (LV) extracted from the above TLE and PP features was explained by these OSTOX and antioxidant biomarkers. The PON1 Q192R genetic variant showed indirect effects on this LV, which were completely mediated by PON1 activity and MDA. Factor analysis showed that a common core could be extracted from TLE, PP, OSTOX and antioxidant scores, indicating that these features are manifestations of a common underlying construct, i.e., a novel pathway phenotype of TLE. Based on the latter, we constructed a new phenotype class that is characterized by increased severity of TLE, PP and OSTOX features and lowered antioxidant defenses. A large part of the variance in episode frequency was explained by increased MDA, lowered antioxidant, and nitric oxide metabolite levels. In conclusion, (a) PP symptoms belong to the TLE phenome, and the signal increased severity; and (b) cumulative effects of aldehyde formation and lowered antioxidants determine epileptogenic kindling.

## 1. Introduction

Epilepsy patients have recurring seizures that are caused by the excessive and synchronized firing of clusters of neurons in the brain [1,2]. Temporal lobe epilepsy (TLE) is the most prevalent kind of epilepsy, occurring at a rate of 10.4 per 100,000 people [1]. Hippocampal sclerosis or mesial temporal sclerosis (MTS) is the most frequent primary epileptic pathology, accounting for 36% of all focal epilepsy pathologies [3,4,5]. Additionally, TLE is associated with a high prevalence of comorbid neuropsychiatric syndromes (approximately 54.1%), particularly depression (42.9%) and anxiety (18.4%), and psychosis, albeit with lower frequency [3,6,7,8]. Most importantly, the high lifetime prevalence of mental disorders in TLE may have a detrimental effect on health-related quality of life [9].

Both oxidative stress toxicity (OSTOX) and a deficiency in antioxidant (ANTIOX) defenses contribute to TLE [10,11,12,13,14]. In TLE, increased ROS production and lipid peroxidation, as well as hippocampal neurodegeneration and reconfiguration of neuronal networks with reactive gliosis contribute to epiloptogenesis and greater sensitivity to subsequent seizures [15]. In a previous study, we found that two OSTOX biomarkers, namely malondialdehyde (MDA, indicating increased lipid peroxidation with aldehyde formation) and advanced protein oxidation products (AOPP, indicating increased protein oxidation), show a substantial area under the receiver operating characteristic (ROC) curve discriminating pure TLE from controls, namely 0.999 for MDA and 0.851 for AOPP [13]. These results suggest that oxidative stress damage plays a significant role in the pathogenesis of TLE/MTS [13]. Both markers indicate several types of oxidative damage that may result in neurotoxicity, thereby contributing to a higher susceptibility of the piriform cortex to seizure-induced injury, apoptotic cell death in hippocampus neuronal cells through activation of the caspase-3 signaling pathway, and increased permeability of the mitochondrial membrane and its enzymes, potentially resulting in neurodegenerative processes [13,16,17].

In individuals with TLE, reduced levels of antioxidants, including superoxide dismutase, paraoxonase 1 (PON1), catalase, glutathione peroxidase, vitamin E, total reactive antioxidant potential (TRAP), and sulfhydryl (-SH) groups, may be observed [13,14,18,19]. Both lowered levels of -SH groups and PON1 activity are highly specific for TLE, resulting in areas under the ROC curves of 0.899 and 0.893, respectively [13,14]. A healthy antioxidant system is critical for preventing lipid peroxidation and protein oxidation, and situations linked with decreased PON1 activity, such as TLE, are associated with an increased risk of developing lipid peroxidation [13,14]. Moreover, PON1 enzymatic activity, which is partly determined by the PON1 Q192R genotype, mediates the effects of this genetic variant on the phenome of TLE [13,20].

There is now evidence that major depression, anxiety disorders such as generalized anxiety disorder (GAD), and schizophrenia are characterized by activated nitro-oxidative stress pathways, including increased lipid peroxidation, as assessed with lipid hydroperoxides (LOOH), MDA, and protein oxidation (increased AOPP), and by lowered antioxidant defenses including lowered PON1 activity, and -SH, TRAP, and glutathione levels [20,21,22,23,24]. Moreover, the depressive, anxiety, and psychotic symptoms due to TLE are strongly associated with the same oxidative stress biomarkers (MDA and AOPP) and antioxidant levels (PON1 activity, -SH groups ad TRAP) [13,14]. These results indicate that lower PON1 activity, which is affected in part by the Q192R genetic variant, and the ensuing damage due to oxidative stress contribute to the pathophysiology of TLE and the psychiatric comorbidities. Nevertheless, there are no data on whether, in TLE, the PON1 status (that is, PON1 activity and the Q192R genetic variant) is associated with increased damage due to oxidative stress and whether these factors may influence the TLE phenome features (including frequency and poor controllability of the seizures, and postictal confusion) and the comorbid psychiatric symptoms. Moreover, there are no data on whether in TLE these biomarkers may predict a common core underpinning both the TLE and psychiatric phenome features.

Recently, novel precision nomothetic methods were developed to construct data-driven, causal disease models [25,26]. To estimate such disease models with structures intended to give causal explanations, we developed a causative framework utilizing causome, protectome, adverse outcome pathways (AOPs), and phenome (descriptive symptomatic assessments) indicators entered in the model as either latent vectors (common cores extracted from a number of interrelated variables) or single indicators [25,26]. We used partial least squares (PLS)-SEM to estimate these machine learning models since PLS-SEM is a causal–predictive technique that places a priority on prediction [27]. As such, we constructed novel precision models that link genetic variants (including the Q192R variant), with AOPs, including oxidative stress and lowered antioxidant defenses, with the phenome of depression and schizophrenia. Moreover, we built (a) endophenotype classes that are groups of patients with a common genetic background and/or AOPs that may affect treatment and prognosis; (b) pathway phenotypes, namely factors extracted from interrelated AOPs and phenome data; and (c) a digital self of each patient that is an ideocratic or personalized profile based on all features (causome, AOP, phenome) of the disorder [27]. Nevertheless, until now, no precision models of TLE and its psychiatric comorbidities have been constructed.

Hence, the current study was conducted to (a) delineate whether the PON1 status is associated with increased oxidative damage and whether these factors may influence the association between TLE and psychiatric comorbidities; (b) construct a precision nomothetic causal disease model, endophenotype classes and pathway phenotypes that may have prognostic and therapeutical consequences for TLE.

## 2. Subjects and Methods

### Participants

We recruited forty healthy controls and 108 TLE patients for this case-control research. Between December 2013 and December 2014, TLE outpatients were admitted to the King Chulalongkorn Memorial Hospital’s Comprehensive Epilepsy Unit in Bangkok, Thailand. All patients were diagnosed with TLE, by a senior neurologist who specializes in epilepsy, based on their clinical features, seizure history, electroencephalogram (EEG) results, and magnetic resonance imaging (MRI). Additionally, the study group of patients with TLE was divided into four subgroups according to the presence of psychiatric comorbidities diagnosed using DSM-IV-TR criteria: (a) pure TLE that is without any comorbidities (*n* = 27); (b) anxiety disorder due to TLE with panic attacks, GAD, or obsessive–compulsive symptoms (*n* = 27); (c) mood disorders due to TLE with depressive features (*n* = 27); and (d) psychotic disorder due to TLE with delusions or hallucinations (*n* = 27).

We excluded TLE patients who met the following criteria: (a) they had DSM-IV-TR axis I diseases other than mood, anxiety, or psychotic disorders due to TLE; (b) they had a recent seizure with or without aura (last week prior to the study), and (c) they had inter-ictal dysphoric disorder (IDD) according to Blumer’s criteria [28]. Additional exclusion criteria for the three comorbidity classes were: (a) the presence of anxiety and psychosis in patients with mood disorders caused by TLE; (b) the presence of mood disorders or psychosis in patients with anxiety disorders caused by TLE; (c) the presence of mood disorders or anxiety in patients with psychotic disorder caused by TLE; and (d) the presence of any psychiatric comorbidity in patients with “pure TLE”. Healthy controls were excluded if they had a diagnosis of epilepsy, including febrile seizures in children, or any DSM-IV-TR axis-1 mental disorder, or if they had a positive family history of epilepsy, mood, or psychotic disorders. Both patients and controls were excluded if they had any of the following: (a) neurodegenerative/neuroinflammatory disorders such as stroke, multiple sclerosis, Huntington’s, Parkinson’s or Alzheimer’s disease; (b) (auto)immune disorders such as psoriasis, diabetes, inflammatory bowel disease, systemic lupus erythematosus, rheumatoid arthritis, or chronic obstructive pulmonary disease; (c) pregnant or lactating women; (d) a medical history of immunomodulatory drug treatment, including glucocorticoids, three months prior to participation in the study, (e) use of therapeutic dosages of antioxidants or ω-polyunsaturated fatty acid supplements; and (f) an inflammatory, immune, or allergic response three months prior to inclusion in the study.

To participate in the research, all subjects supplied written informed consent. The study was approved by the Institutional Review Board of Chulalongkorn University’s Faculty of Medicine in Bangkok, Thailand (IRB number 305/56), in accordance with the International Guideline for the Protection of Human Subjects as required by the Declaration of Helsinki, The Belmont Report, the CIOMS Guideline, and the International Conference on Harmonization on Good Clinical Practice.

## 3. Measurements

### 3.1. Clinical

A senior neurologist and an epilepsy-specialized senior psychiatrist performed semi-structured interviews. The neurologist collected sociodemographic data and TLE features, including family history of epilepsy, age at onset of TLE, type of epilepsy, location of the lesion, seizure frequency, history of post-ictal confusion, antiepileptic drug (AED) use, seizure control (seizure-free, fairly controlled, and poorly controlled seizures), seizure type, and history of aura. TLE was diagnosed based on a history of partial seizures and electroencephalogram (EEG) recordings of epileptiform activity in one or both temporal regions. The senior psychiatrist examined the TLE patients and controls for psychotic symptoms, anxiety, and depression using DSM-IV-TR criteria. TLE-related mood disorders include significant depression during an acute episode or in partial remission, as well as ictus-related depression. Patients with panic attacks, GAD, obsessive–compulsive symptoms, and ictus-related anxiety such as terror and horror are included in the diagnostic of anxiety disorder caused by TLE. Psychotic illness associated with TLE include delusions (persecution, paranoid; possessed), hallucinations (taste, visual, auditory, and olfactory), and ictus-related psychoses. These psychoses may be classified as ictal, pre-ictal, post-ictal, psychotic aura, peri-ictal, interictal, or schizophrenic-like. Nevertheless, fear, déjà vu, déjà vecu, forced thoughts, horror and out-of-body experiences were not considered psychotic in this study. Additionally, the senior psychiatrist (BK) examined controls and TLE patients using the Brief Psychiatric Rating Scale (BPRS), the Hamilton Depression (HDRS) and Anxiety (HAM-A) Rating Scales, as well as the Mini Mental State Examination (MMSE) [29,30,31,32]. Using the scores of the BPRS, HDRS, and HAM-A, we have computed two additional symptom scores, namely psychosis: computed as the sum of conceptual disorganization (item 4 BPRS) + suspiciousness (item 11 BPRS) + hallucinations (item 12 BPRS) + unusual thought content (item 15 BPRS); and a general psychopathology score (z total PP) computed as the sum of the z values of the BPRS (z BPRS) + z HAM-D + z HAM-A [13,14].

The body mass index was computed as the ratio of weight in kilograms to height in meters^2^, and tobacco use disorder (TUD) was diagnosed using DSM-IV-TR criteria.

### 3.2. Assays

After an overnight fast, blood was drawn for the biomarker assays at 8:00 a.m. Aliquots of serum were prepared and kept at −80 °C until thawed for assay. PON1 status, AOPP, LOOH, MDA, NOx, TRAP, and -SH groups were measured. The approaches have been previously reported [13,14]. LOOH was measured by chemiluminescence in the dark, at 30 °C for 60 min, using a Glomax Luminometer (TD 20/20) [33,34]. The findings are reported in relative light units. MDA levels were determined using high-performance liquid chromatography (HPLC Alliance e2695, Waters, Barueri, SP, Brazil) in complexation with two molecules of thiobarbituric acid [35]. The following conditions were used: a column Eclipse XDB-C18 (Agilent, Santa Clara, CA, USA); a mobile phase composed of 65 percent potassium phosphate buffer (50 mM pH 7.0) and 35% HPLC grade methanol; a flow rate of 1.0 mL/minute; a temperature of 30 °C; and a wavelength of 532 nm. The MDA content of the samples was determined using a calibration curve and is given in mmol of MDA/mg proteins. AOPP was measured using a microplate reader (EnSpire^®^, Perkin Elmer, Waltham, MA, USA) and is expressed in mM of equivalent chloramine T [36,37]. The NO metabolites (NOx) were quantified using a microplate reader (EnSpire^®^, Perkin Elmer, Waltham, MA, USA) at a wavelength of 540 nm by detecting the concentrations of nitrite and nitrate [38]. Results are represented as M. Because the PON1 polymorphism causes variations in hydrolysis capability, it is possible to stratify genotypes after phenotypic measurement of enzyme activity. We employed 4-(chloromethyl)phenyl acetate (CMPA) (CMPA, Sigma, Burlington, MA, USA), which is an alternative to the use of the toxic paraoxon, and phenylacetate (Sigma, Burlington, MA, USA), under high salt conditions, to stratify the functional genotypes of the PON1Q192R polymorphism, namely Q/Q, Q/R, and R/R [14,20]. PON1 activities were determined by the rate of hydrolysis of CMPA, which is influenced by the Q192R polymorphism, and phenylacetate under low salt condition (AREase), which is less influenced by the Q192R polymorphism. The rate of hydrolysis was assessed using a Perkin Elmer^®^ EnSpire model microplate reader (EnSpire, Waltham, MA, USA) at a wavelength of 270 nm during a 4 min period (16 readings with a 15 s interval between readings) while maintaining a temperature of 25 °C [14,39] The activity was quantified in units per milliliter (U/mL). The PON1 Q192R polymorphism influences the activity of the PON1 enzymes, thereby altering their ability to prevent lipid oxidation [20,40], but the direction of this change is substrate dependent [41,42]. RR homozygotes show a greater efficacy detoxifying substrates, including paraoxon, CMPA and 5-thiobutil butyrolactone (TBBL) [39,43,44], although the influence on TBBL (i.e., lactonase activity) is lower (30–50% higher in RR) than paraoxon (100–200% higher in RR) [43]. The Q allozyme is more efficient in detoxifying substrates such as diazoxon. In the present study, we report on CMPAase activity [14]. The -SH groups were determined using a microplate reader (EnSpire^®^, Perkin Elmer, Waltham, MA, USA) at a wavelength of 412 nm, and the findings are reported in M [14]. TRAP was determined using a microplate reader (Victor X-3, Perkin Elmer, Waltham, MA, USA), and the findings are represented in M Trolox [45]. We assessed the thiol or sulfhydryl (-SH) groups using a microplate reader (EnSpire^®^, Perkin Elmer, Waltham, MA, USA) at a wavelength of 412 nm, and results are expressed in μM [46,47].

Based on these results, we generated three z unit-weighted composite scores: (a) an indicator of oxidative stress toxicity (OSTOX) defined as z transformation of MDA (z MDA + z LOOH + z AOPP; (b) an index of antioxidant activities (ANTIOX) defined as z TRAP + z thiol (-SH) groups + z CMPAase activity; and (c) the OSTOX/ANTIOX ratio defined as z OSTOX—z ANTIOX [13,14].

## 4. Statistics

The analysis of variance was used to compare continuous variables between groups, while the analysis of contingency tables (X^2^-tests) was used to compare nominal variables. To account for multiple statistical tests, we used p-corrections for false discovery rate (FDR) [48]. Multivariate GLM analysis was employed to examine the effects of explanatory variables (age, sex, BMI, drug state, smoking) on a set of dependent variables (phenome and biomarker scores). Consequently, we also examined the tests for between-subject effects. Multiple regression analysis (automatic, stepwise) was used to denote the biomarkers that are significantly linked with the clinical outcome data. VIF and tolerance values were used to evaluate for multicollinearity in all regression analyses. In addition, we investigated changes in R^2^, multivariate normality (using Cook’s distance and leverage), and homoscedasticity (using White and modified Breusch–Pagan tests for homoscedasticity). We utilized an automated stepwise (step-up) technique with *p*-to-enter and *p*-to-remove values of 0.05 and 0.06, respectively. Additionally, all findings were bootstrapped using five thousand samples, and the bootstrapped results are shown in the event of discrepancies. All tests were two-tailed, and statistical significance was defined as a *p* value of 0.05. Exploratory factor analysis and principal component (PC) model was performed, and the Kaiser–Meier–Olkin (KMO) sample adequacy metric was assessed to determine factorability (considered satisfactory when >0.7). The first PC was a general construct underpinning the input variables when all loadings were >0.6 and the variance explained by the first PC was >50.0 percent. Replacement of missing values was performed using the series mean method only when there were <10% missing values. In the total data file, there were only six missing values in frequency and uncontrollability of seizures (*n* = 6). All biomarker data were complete in all participants. The data were analyzed using IBM SPSS28 for Windows. Moreover, confirmatory factor analysis (CFA) was used to examine the loadings of the outer model in partial least squares (PLS) analysis (using the factor weighting scheme), and factors were considered to have sufficient construct validity and convergent validity when all loadings were >0.6 at *p* < 0.001, rho_A was >0.7, and the average variance extracted >0.5.

PLS-SEM path analysis is a statistical approach for predicting complex cause–effect linkages utilizing both single indicators (variables) and latent variables (factors based on a vector of highly connected indicators) [49,50]. Without imposing distributional assumptions on the data, PLS allows for the estimation of complex multi-step mediation models with several latent constructs, indicator variables, and structural pathways (associations between indicators or latent vectors). PLS Smart was used to analyze the multi-step, multiple mediation relationships between input factors (Q192R PON1 genotype, CMPAase activity, other antioxidants, oxidative stress biomarkers) and the output variables, namely severity of epilepsy and comorbid psychiatric disease. According to the power analysis, the predicted sample size should be at least 134 to obtain a power of 0.8, an effect size of 0.1, and an alpha of 0.05 in a multiple regression (or PLS) analysis with up to six explanatory variables. Each input variable was entered as a single indicator (e.g., sex, age, Q192R PON1 genotype models), but the output variables (clinical data) were entered as latent vectors (reflective models) constructed from several indicators. We investigated whether it was possible to combine the psychiatric (BPRS, HDRS, HAM-A) and epilepsy data (frequency, controllability, history of aura, postictal confusion) using first PCA and consequently CFA as explained above. Complete path analysis with 5000 bootstrap samples (and mean value replacement) was performed only when the outer and inner models matched the following quality criteria: (a) model quality as measured by the SRMR index is less than 0.08; (b) outer model loadings on the latent vectors exhibit accurate construct validity and convergent validity as explained above; (c) the latent vectors are not mis-specified as a reflective model as probed with confirmatory tetrad analysis (CTA); (d) the model and variables show adequate predictive performance as established using PLS Predict with a 10-fold cross-validation and blindfolding; and (e) compositional invariance is established employing predicted–oriented segmentation analysis, multi-group analysis, and measurement invariance assessment. Using complete PLS analysis (two-tailed) on 5000 bootstrap samples (bias-corrected and accelerated bootstrap), we then computed pathway coefficients with exact *p* values, total and specific indirect effects, and total effects.

## 5. Results

### 5.1. First Precision Nomothetic Model

Table 1 shows a first factor (FA1) extracted from frequency of seizures, a history of aura, controllability of the seizures, the diagnosis of TLE, and postictal confusion (named: TLE phenome). Using PC analysis (PCA), the first PC explained 66.6% of the variance and had loadings that were all >0.6 (factorability was adequate). CFA showed similar loadings and that the convergent validity (as assessed with AVE values) and construct reliability (as assessed with rho_A) were more than adequate. Table 1 shows the features of a second constructed factor (FA2) that combines the five TLE features coupled with four psychiatric features (HDRS, HAM-A, BPRS and z PP scores; named: TLE-PP phenome). Using PCA, all loadings were >0.6, the first PC explained 61.3% of the variance, and the analysis showed adequate factorability. CFA showed that the convergent validity and construct reliability were more than adequate. As such, this latent vector is a common core underpinning all manifestations of the TLE-PP phenome.

Figure 1 shows a first causal model with the combined TLE and psychiatric data (see FA2, named TLE-PP phenome) as output variables and the Q192R variant (additive model), CMPAase activity, -SH groups and MDA as input variables, whereby MDA functions as a mediator between the effects of CMPAase on the TLE-PP phenome. Moreover, we also considered that MDA may mediate the effects of the path from -SH groups to CMPAase on the TLE-PP latent vector. The results of the PLS path analysis on 5000 bootstrap samples after feature selection, multi-group analysis, PLS predict analysis, and prediction-oriented segmentation are shown in Figure 1. With SRMR = 0.049, the model’s overall fit was acceptable. Additionally, the construct reliability of the TLE-PP latent vector was satisfactory as shown in Figure 1, while this factor was not misspecified as a reflective model (results of CTA), and the construct cross-validated redundancy was adequate (0.287). Smart Predict showed that all Q^2^ values were positive, indicating that they outperformed the naive benchmark. We observed that 57.2% of the variation in the TLE-PP phenome could be explained by the regression on MDA and CMPAase and a moderation (interaction) effect between MDA and CMPAase. Moreover, 34.9% of the variance in MDA was explained by the regression on CMPAase and -SH groups, and 27.6% of the variance in CMPAase was explained by the Q192R genotype and -SH groups combined.

Here, there were no total (that is direct and mediated) effects (*t* = 1.47, *p* = 0.142) and no direct effects (pathway coefficient *pc* = 0.133, *p* = 0.254) of the Q192R genotype on the TLE-PP phenome, while the Q192R genotype has significant total indirect effects on the TLE-PP phenome (*t* = 3.21, *p* = 0.001), which are mediated by CMPAase (*t* = 2.02, *p* = 0.043) and the path from CMPAase to MDA (*t* = 2.67, *p* = 0.008). Moreover, there were highly significant total effects of -SH groups (*t* = −5.62, *p* < 0.001) on the TLE-PP phenome that were mediated by MDA (*t* = −3.48, *p* = 0.001) and the path from CMPAase to MDA (*t* = 3.41, *p* = 0.001). Moreover, we found a significant interaction (moderating) effect between CMPAase and MDA on the phenome. Overall, the TLE-PP phenome is predicted by CMPAase and MDA and an interaction between CMPAase and MDA, with significant indirect effects of Q192R genotype and -SH groups. As such, there are significant total indirect effects of CMPAase (*t* = −4.11, *p* < 0.001) on the TLE-PP phenome.

### 5.2. Prediction of the Phenome on TLE-PP Using Biomarkers

Table 2, model #1 shows that 50.4% of the variance in the TLE-PP score could be explained by the regression on ANTIOX and age (both inversely associated) and OSTOX (positively associated). Figure 2 displays the partial regressions of the TLE-PP score on ANTIOX (after considering the effects of the other variables listed in model #1). In Table 2, Model #2, we entered the single biomarkers and observed that 51.1% of the variance in the TLE-PP score could be explained by the regression on MDA (positively associated), CMPAase, TRAP and NOx (all inversely associated). Figure 3 shows the partial regression of TLE-PP score on MDA (after considering the effects of the variables listed in model #2).

Table 2, model #3 shows that 31.8% of the variance in the frequency of seizures was explained by MDA (positively), NOx, ANTIOX and age (all inversely). Table 2, model #4 shows that 23.8% in the uncontrollability of seizures was explained by ANTIOX and NOx (both inversely) and MDA (positively).

### 5.3. Precision Nomothetic Model 2

Figure 4 shows a second PLS model with a latent vector extracted from TLE, PP, MDA, and OSTOX data as output variable (TLE-PP-OS) and TRAP, -SH groups, CMPAase and the PON1 Q192R genotype as explanatory variables. With SRMR = 0.050, the overall fit of the model was acceptable, and the construct reliability and convergent reliabilities of the latent vector were accurate, as shown in Figure 2 and by adequate rho_A (0.934) and AVE (0.546) values. Moreover, the construct cross-validated redundancy was adequate (0.225), and this factor was not misspecified as a reflective model (results of CTA). Smart Predict showed that all Q^2^ values were positive, indicating that they outperformed the naive benchmark. In this model, there were highly significant total effects of the PON1 Q192R variant (*t* = 3.99, *p* < 0.001) and -SH groups (*t* = −6.25, *p* < 0.001) on the TLE-PP-OS pathway phenotype. The effects of the genetic variant were mediated by CMPAase (*t* = 3.69, *p* < 0.001) and the path from CMPAase to TRAP (*t* = 3.19, *p* = 0.002). The indirect effects of -SH groups on the phoneme were mediated by CMPAase (*t* = −4.48, *p* < 0.001) and the path from CMPAase to TRAP (*t* = 2.11, *p* = 0.035). We found that 43.6 percent of the variance in the TLE-PP-OS phenome can be explained by the regression on CMPAase, TRAP and -SH groups, 9.1 percent of the variance of TRAP is explained by CMPAase, and 28.7 percent of the variance in CMPAase by the Q192R genetic variant and -SH groups.

### 5.4. Construction of Endophenotype Classes and Pathway Phenotypes

Since the antioxidants TRAP, -SH and CMPAase also have strong effects on the TLE-PP-OS pathway phenotype, we examined whether ANTIOX could be successfully added to this endophenotype. Table 1 (FA3) shows that the loadings on the first PC and first factor extracted from the five TLE, four PP and MDA, OSTOX and ANTIOX (named: TLE-PP-OS-AO) were all >0.6 and that this first PC explained 58.0% of the variance with adequate factoriability, while the first factor showed adequate rho_A and AVE values.

Based on the latent variable scores of the TLE-PP-OS-AO pathway phenotype, we used a visual binning method (examination of the apparent modes and local minima in the frequency histogram) to divide the study sample into three non-overlapping samples (the cutoff points were −0.53 and 0.68, respectively). This separation into three groups (namely, healthy controls and two TLE groups) showed an adequate silhouette measure of cohesion and separation of 0.8 (based on Akaike’s information criterion). The features of these new endophenotype classes are shown in Table 3. The latent vector scores extracted from the TLE and TLE-PP scores were significantly different between the three groups. The high TLE-PP-OS-AO group was characterized by a high number of seizures, postictal confusion, poorer controllability of the seizures, increased depression, anxiety, psychosis, and z PP scores; lower CMPAase activity and -SH concentrations; and increased MDA and OSTOX/ANTIOX levels. As such, we have constructed a new endophenotype class characterized by interrelated increments in TLE and PP phenome, and OSOTOX/ANTIOX scores.

Using multivariate GLM analysis with TLE, PP, TLE-PP, TLE-PP-OS, OSTOX, and ANTIOX as dependent variables we examined the effects of age, sex, BMI, smoking and the drug state of the participants. Some patients were treated with carbamazepine (*n* = 61), phenytoin (*n* = 38), lamotrigine (*n* = 27), valproate (*n* = 34), levetiracetam (*n* = 38), phenobarbital (*n* = 26), clonazepam (*n* = 10), topiramate (*n* = 12), clobazam (*n* = 58), gabapentin (*n* = 8), CaCO_3_ (*n* = 13), folic acid (*n* = 27), antidepressants (*n* = 16), antipsychotics (*n* = 9), and anxiolytics (*n* = 10). We could not detect significant effects of these explanatory variables either in the multivariate GLM analysis or in tests for between-subject analysis even without p-correction for multiple testing.

## 6. Discussion

### 6.1. A Common Core Underpins TLE and Comorbid Psychopathology (PP)

The study’s first major discovery is that a common core (with adequate validity reliability) underlays the features of TLE (including frequency of seizures, history of aura and postictal confusion, and poorer seizure controllability) and severity of psychopathology including depression, anxiety, and psychosis. The data demonstrate that PP symptoms are prominent features of the same latent vector (the phenome of TLE-PP) that should be considered as the source of the diverse clinical manifestations. The presence of more severe PP symptoms signals increased severity of TLE, including frequency and uncontrollability of the seizures. Importantly, in mood disorders, there is an association between increasing staging (frequency of depressive and hypomanic episodes and suicidal attempts) and the severity of these illnesses [26,51]. Here, (basolateral amygdala) kindling is a typical paradigm for the development of epileptic seizures in which the length and behavioral involvement of the produced seizures increase as seizures reoccur [52]. As such, a seizure may increase the risk of further seizures (“seizures beget seizures”). Likewise, the kindling theory of mood disorders considers that untreated mood disorders tend to progress, and that—with repetition—the episodes become sensitized and more autonomous with a shorter interval [53]. Thus, both TLE and mood disorders are characterized by an increased vulnerability to episode reoccurrence with shorter intervals as a function of the number of prior episodes.

Moreover, in other neurological and medical disorders, the severity of affective symptoms is part of a common core underpinning the illness. For example, in schizophrenia, a common latent vector may be extracted from psychotic and affective symptoms, including depression, anxiety, and hypomania [54]. In stroke, a common factor may be extracted from the disabilities as assessed with the NIHSS score and affective symptoms, and this vector is strongly predicted by the volume and location of the acute lesions due to stroke, white matter hyperintensities, and hypertension [55]. Depression severity belongs to the same common core (a validated latent vector) that includes clinical features of atherosclerosis and unstable angina, class III/IV unstable angina, increased atherogenicity, and insulin resistance [56]. Our findings that PP symptoms are a key part of the same clinical core that underpins the manifestations of medical disorders, such as TLE, are at odds with the view that depression due to medical disorders is explained by a number of psychosocial stressors associated with the illness, such as the personal meaning given to the functional losses, views and beliefs about the illness itself, personality features, coping mechanisms, social support, and the life stage [57]. The view that such stressors exert a load on the mind–brain link producing affective symptoms is a folk psychology-like explanation [58]. As such, psychosocial psychiatry attributes behavioral symptoms including depression, and anxiety as a reaction to beliefs and perceptions [58]. Treatment plans for depression and anxiety due to medical diseases including epilepsy are then devised based on these folk-like theories, and these include specialized cognitive or dynamic behavioral psychotherapies, advice, education, problem solving, interpersonal therapy, and treatment with antidepressants [57,58]. However, our results show that TLE and PP are clinical manifestations of shared pathways.

### 6.2. Oxidative Stress and the TLE-PP Common Core

The second major finding of this study is that 57.2% of the severity in the TLE-PP phenome is strongly predicted by the combined effects of increased OSTOX (especially MDA and AOPP) and lowered ANTIOX (CMPAase, -SH groups, and TRAP) levels. These findings extend those of previous reports that TLE [13,14,18,19,59], affective disorders [20,21,22,24], and schizophrenia [20,60] are characterized by indicants of oxidative damage.

Here, we show that there are no direct associations between the Q192R gene variant and the TLE-PP phenome, although there were significant indirect effects of the additive PON1 model, which were completely mediated by the effects of CMPAase on MDA and, consequently, the phenome. As such, CMPAase is a successful mediator. This is an example of “indirect-only mediation”, whereby the mediator (PON1 activity) mediates the association between the gene variant and the phenome even when there are no direct (association between PON1 gene and phenome) and total (direct + indirect) effects [61]. The latter should not be employed as a gatekeeper to examine mediation effects [62,63].

We would not have discovered the PON1 gene variant’s impact if we had not measured the PON1 gene products. Many genetic association studies are likely to yield false negative results in the case of “indirect-only mediation” effects. Many positive association studies, however, may be based on spurious associations (false positives) because the proteins that mediate the effects of a specific gene may interact with many other proteins/enzymes (in protein–protein interaction (PPI) networks) that predict the outcome variable, implying that it is the interactors, not the gene product, that predict the outcome. Furthermore, some interacting proteins may counteract with the gene-product functions, resulting in false negative results.

Our model also showed that the effects of CMPAase on the TLE-PP phenome are partly mediated by MDA in a distal mediated model, namely the path coefficient from MDA to the phenome is greater than that from CMPAase to MDA. This is an example of complementary mediation whereby both the indirect and the direct paths are significant and point in the same direction [61]. A significant direct effect suggests that the theoretical framework may be incomplete, and it guides the examination of additional paths that match the direct effects’ signs of a direct effect. As a result, we discovered that an interaction between CMPAase (the moderator) and MDA further explained part of the variance in the TLE-PP phenome.

Nevertheless, our model became even more complex after considering the effects of -SH groups on PON1 activity and considering that -SH groups, including these in PON1, may contribute to the prediction of the TLE-PP-OS phenome. Protein thiols are often positively associated with PON1 activity [64] and are part of TRAP, while PON1 activity is often associated with TRAP levels [65,66]. PON1 activity is protected by antioxidants, and thiol groups may be regenerated through reductive recycling by cell reductants, including the glutathione system [67]. Our PLS analysis revealed that these causal links from -SH groups to CPMAase activity are significant, whereby part of the effects of -SH groups on the phenome are mediated by CMPAase, and part of the effects of the latter are mediated by TRAP, thereby establishing a multi-step, mediating model. By inference, the lowered CMPAase activity in TLE is partly determined by the Q192R variant and lowered thiol groups. PON1 contains three cysteine residues that have functional properties (formation of disulfide bonds and high affinity metal binding), while one of these (position 284) is a free thiol that, at least in part, determines PON’s antioxidant capacities against LDL oxidation [68].

Not only PON1 activity [20,65,69] but also low and high molecular weight thiols, both cellular and in serum, may protect against oxidation of LDL and oxidative damage [70]. Therefore, lowered levels of CMPAase, -SH groups and TRAP may increase the vulnerability to lipid peroxidation and thus aldehyde formation and protein oxidation [21], thereby explaining the strong effects of those antioxidants on the TLE-PP-OS phenome. Conversely, PON1 activity may be attenuated by oxidative stress, including the effects of myeloperoxidase, peroxides, hydroperoxides, oxidized low density lipoprotein and S-nitrosylation through modifications of the free thiol group when the latter is consumed by preventing oxidative damage [20,67,68,69]. Phrased differently, the results show that lowered antioxidant potential, which is in part determined by the Q192R PON1 variants and lowered -SH groups, coupled with increased aldehyde formation are a key part of the TLE-PP phenome of epilepsy. This is further underscored by our findings that a common core may be extracted from the TLE and PP phenome, OSTOX and ANTIOX and thus that those constructs are all strongly interrelated manifestations of the same core.

Overall, our findings indicate that decreased CMPAase and -SH groups, as well as enhanced aldehyde production, are potential therapeutic targets for treating TLE. Moreover, lowered NOx levels (indicating nitric oxide production) are associated with the TLE-PP phenome and the kindling of seizures, suggesting increased NO consumption through hypernitrosylation that may further enhance inflammatory reactions and oxidative stress as well as neurodegenerative processes [71]. In this regard, increased neuronal NO synthase activity is involved in kindling, oxidative damage, and endoplasmatic stress [72], and S-nitrosylation is associated with affective disorders [73].

### 6.3. A Novel Pathway Phenotype and Endophenotype Class

The current study’s third finding is the construction of two novel precision medicine concepts in TLE: (a) a new pathway phenotype (the TLE-PP-OS-AO construct), which integrates severity of TLE, psychiatric comorbidities, and AOPs; and (b) a new endophenotype class characterized by highly increased TLE features such as number of episodes, worse seizure controllability, PP scores, OSTOX, and lowered ANTIOX defenses. Our findings show that this new class of patients is a more severe group, and that there is a gradual increase in “epileptogenic kindling” and uncontrollability of the seizures along a severity of illness determined by the cumulative effects of aldehyde production and decreased antioxidant defenses. The reoccurrence or staging of mood disorders is linked to reduced CMPAase activity and oxidative stress indicators [27,45].

## 7. Limitations

This study would have been more interesting if we had included other oxidative stress biomarkers, including superoxide dismutase, catalase, and glutathione, myeloperoxidase, 4-hydroxy-nonenal, and indicants of nitrosative stress. All results of this study were controlled for effects of age, sex, BMI, smoking and the drug state of the patients. We could not find any significant effects of the drug state of the patients (and age, sex, BMI, smoking) on any of the phenome scores and pathway endophenotypes constructed in the current study. Previously we have shown and discussed that the use of AEDs does not affect the PP phenome or any of the OSTOX and ANTIOX biomarkers. Previous reports also could not find significant effects of AEDs on MDA, protein carbonyls, NO levels, and antioxidant enzymes [13,14,74].

## 8. Conclusions

TLE, PP, OSTOX, and ANTIOX levels have a shared core, suggesting that these characteristics are expressions of a single underlying construct, i.e., a new pathway phenotype of TLE. We created a new endophenotypic class that is characterized by increased severity of TLE, PP, OSTOX and ANTIOX characteristics. The results show that PP symptoms are part of the TLE phenome and signal greater severity. Epileptogenic kindling is at least in part determined by the cumulative effects of increased aldehyde production and reduced antioxidant defenses.

Overall, the severity of the new pathway phenotype of TLE and allocation to the new endophenotype class PP-OS-AO, characterized by increased psychopathology and oxidative damage and lowered antioxidant defenses, signal greater severity of TLE, and increased frequency and uncontrollability of seizures. In animal models, there is some evidence that administration of antioxidants, including acetyl-carnitine, L-carnitine, Ginkgo biloba leaf extract, Lantana camara and vanillic acid, exert neuroprotective, anticonvulsant and/or anti-kindling effects [75,76,77,78,79]. There is also some evidence that in TLE patients, treatment with antioxidants such as selenium and vitamin E and C may have some efficacy [80]. Nevertheless, in preclinical models, natural polyphenols have mixed affects, and N-acetylcysteine may aggravate seizures and improve depressive-like behaviors [81,82]. The results of the current precision medicine study show that PON1 activity and aldehyde formation may be the most appropriate drug targets to treat severe TLE. Future research should develop and trial new drugs targeting the PON1–aldehyde pathway.

## Figures and Tables

**Figure 1 antioxidants-11-00803-f001:**
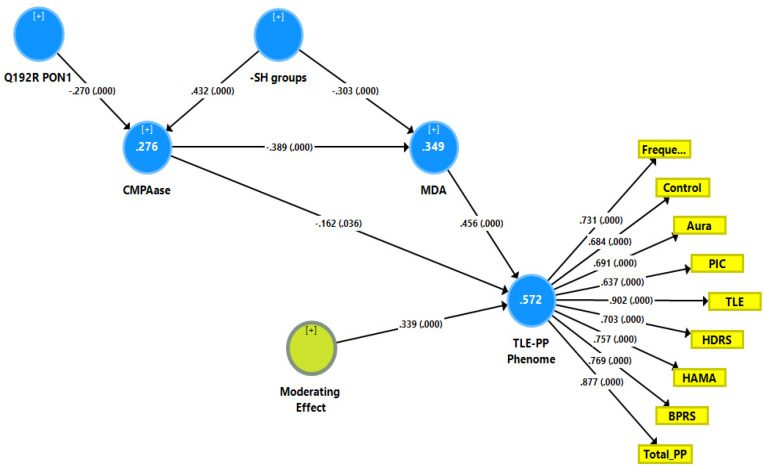
Results of partial least squares (PLS) showing a first causal model with the combined temporal lobe epilepsy (TLE) and psychopathology (PP) score as output variable (TLE-PP phenome) and the Q192R variant (additive model), CMPAase 4-(chloromethyl)phenyl acetate ase; -SH groups and malondialdehyde (MDA) as input variables, whereby CMPAase and MDA mediate the effects of the genetic variant on the TLE-PP phenome. Freque: frequency of seizures; control: uncontrollability of seizures; aura: history of aura; PIC: post-ictal confusion; HDRS and HAM-A: Hamilton Depression and Anxiety Rating Scales; BPRS: Brief Psychiatric Rating Scale; PP: psychopathology.

**Figure 2 antioxidants-11-00803-f002:**
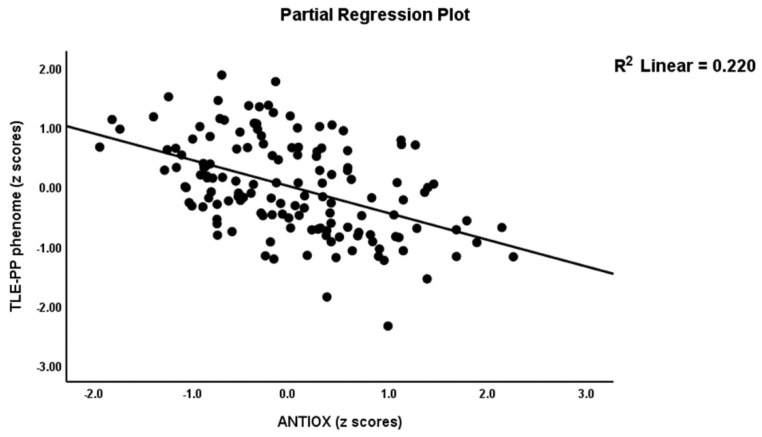
Partial regression of temporal lobe epilepsy—psychopathology phenome score on the index of antioxidant defenses (ANTIOX).

**Figure 3 antioxidants-11-00803-f003:**
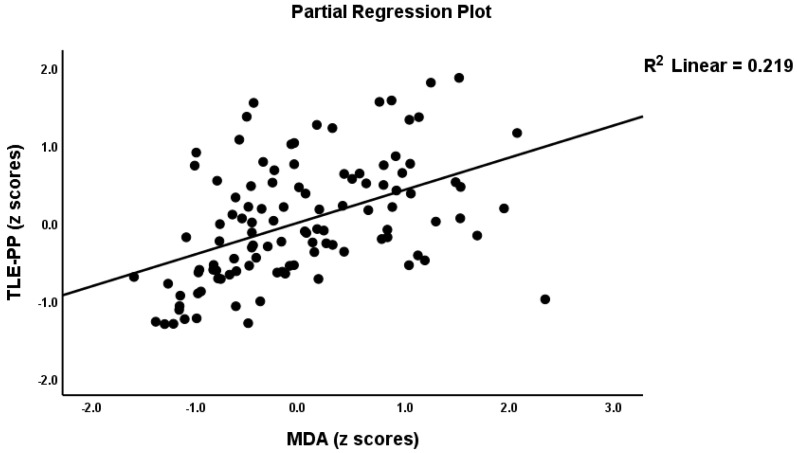
Partial regression of the temporal lobe epilepsy—psychopathology phenome score on malondialdehyde (MDA).

**Figure 4 antioxidants-11-00803-f004:**
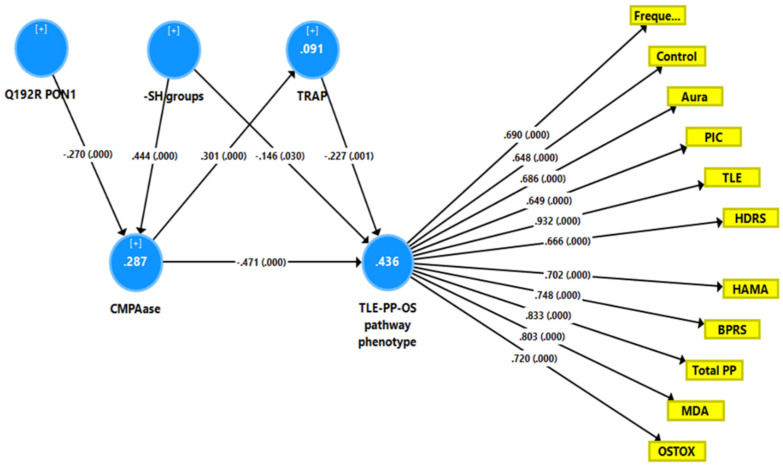
Results of partial least squares (PLS) analysis showing a second causal model with the combined temporal lobe epilepsy (TLE), psychopathology (PP) and oxidative stress (OS) score as output variable (TLE-PP-OS pathway phenotype) and the Q192R variant (additive model), CMPAase 4-(chloromethyl)phenyl acetate ase, -SH groups, and total radical trapping parameter (TRAP) as input variables, whereby TRAP and CMPAase activity mediate the effects of the genotype and -SH groups on the TLE-PP-OS pathway phenotype. OSTOX: index of oxidative toxicity; MDA: malondialdehyde; Freque: frequency of seizures; control: uncontrollability of seizures; aura: history of aura; PIC: post-ictal confusion; HDRS and HAM-A: Hamilton Depression and Anxiety Rating Scales; BPRS: Brief Psychiatric Rating Scale; PP: psychopathology.

**Table 1 antioxidants-11-00803-t001:** Results of principal components (PCA) and confirmatory factor (CFA) analysis.

Variables	Loadings		
	FA1:TLE	FA2:TLE-PP	FA3:TLE-PP-OS-AO
Frequency insults	0.868/0.790	0.767/0.732	0.734/0.648
Controllability	0.821/0.767	0.744/0.685	0.699/0.618
History of aura	0.758/0.756	0.693/0.687	0.708/0.681
History of postictal confusion	0.743/0.709	0.685/0.634	0.706/0.633
Temporal lobe epilepsy	0.889/0.897	0.899/0.900	0.943/0.938
HDRS	-	0.745/0.707	0.701/0.648
HAMA	-	0.812/0.761	0.743/0.678
BPRS	-	0.772/0.770	0.755/0.743
Total Psychopathology		0.913/0.880	0.862/0.813
Malondialdehyde	-	-	0.816/0.805
OSTOX	-	-	0.715/0.732
ANTIOX	-	-	−0.742/0.730
KMO	0.757	0.842	0.771
Bartlett	346.35 (*df* = 10) *	1031.94 (*df* = 36) *	1336.00 (*df* = 66) *
% variance explained	66.9%	61.6%	58.3%
Rho_A	0.891	0.929	0.949

* All *p* < 0.001. Hamilton Depression (HDRS) and Anxiety (HAM-A) Rating Scales; Brief Psychiatric Rating Scale (BPRS); OSTOX: index of oxidative stress toxicity; ANTIOX: index of antioxidant capacity, AVE: average variance explained. x/y: loadings obtained in PCA and CFA, respectively.

**Table 2 antioxidants-11-00803-t002:** Results of multiple regression analyses, with the phenome score or temporal lobe epilepsy features as dependent variables and biomarkers as explanatory variables.

	Explanatory Variables	*β*	*t*	*p*	F_model_	*df*	*p*	R^2^
TLE-PP score	**Model #1**				47.47	3/139	<0.001	0.504
	OSTOX	0.365	5.11	<0.001				
	ANTIOX	−0.449	−6.29	<0.001				
	Age	−0.138	−2.30	0.023				
TLE-PP score	**Model #2**				36.02	4/138	<0.001	0.511
	MDA	0.415	5.72	<0.001				
	CMPAase	−0.298	−4.10	<0.001				
	TRAP	−0.143	−2.24	0.027				
	NOx	−0.124	−2.04	0.043				
Frequency seizures	**Model #3**				16.45	4/141	<0.001	0.318
	MDA	0.350	3.89	<0.001				
	NOx	−0.158	−2.21	0.028				
	ANTIOX	−0.210	−2.35	0.020				
	Age	−0.185	−2.63	0.010				
Uncontrollability seizures	**Model #4**				14.69	3/141	<0.001	0.238
	ANTIOX	−0.224	−2.38	0.019				
	MDA	0.242	2.55	0.012				
	NOx	−0.224	−2.98	0.003				

TLE-PP: phenotype score extracted from frequency of seizures, controllability of seizures, a history of aura, postictal confusion, temporal lobe epilepsy, the severity of depression, anxiety, and general psychopathology; OSTOX: composite score reflecting oxidative toxicity; ANTIOX: composite score reflecting antioxidant defenses; MDA: malondialdehyde; CMPA: 4-(chloromethyl)phenyl acetate; TRAP: total radical trapping parameter; NOx: nitric oxide metabolites; HAM-A: Hamilton Depression Anxiety score.

**Table 3 antioxidants-11-00803-t003:** Differences in socio-demographic, clinical and biomarker data between healthy controls (HC) and patients divided according to the factor analysis score (FA3) shown in Table 1 into those with low and higher pathway phenotype (TLE-PP-OS-AO) scores.

Variables	HC ^a^ *n* = 40	Low TLE-PP-OS-AO ^b^ *n* = 56	High TLE-PP-OS AO ^c^ *n* = 52	F/X^2^	*df*	*p*
TLE phenome	−1.309 (0.328) ^b,c^	0.188 (0.562) ^a,c^	0.805 (0.509) ^a,b^	217.20	2/145	<0.001
TLE-PP Phenome	−1.482 (0.098) ^b,c^	0.257 (0.418) ^a,c^	0.863 (0.390) ^a,b^	532.26	2/145	<0.001
Age (years)	37.4 (12.8)	39.1 (10.7)	37.8 (10.3)	0.30	2/145	0.735
Male/Female	10/30	19/37	16/36	0.88	2	0.643
BMI (kg/m^2^)	24.0 (4.3)	23.3 (3.8)	23.6 (4.3)	0.33	2/142	0.720
Lives alone (No/Yes)	14/26	18/35	14/38	0.87	2	0.646
Education (years)	14.2 (4.9) ^b,c^	10.9 (4.5) ^a^	10.0 (5.0) ^a^	9.51	2/143	0.001
TUD (N/Y)	38/2	47/7	44/8	2.52	2	0.284
Number of seizures *	-	7.2 (14.1) ^c^	26.8 (66.3) ^b^	9.7	1/106	0.002
Postictal confusion (No/Yes)	-	27/26	11/40	9.67	1	0.002
Aura (No/Yes)		16/38	10/42	1.55	1	0.214
Controllability seizures (free/fair/poor)	-	18/11/8	6/14/18	10.20	2	0.006
BPRS *	18.3 (1.1) ^b,c^	28.8 (7.3) ^a,c^	34.9 (8.4) ^a,b^	68.97	2/143	<0.001
HDRS *	0.6 (2.0) ^b,c^	8.0 (5.0) ^a,c^	12.0 (7.9) ^a,b^	45.28	2/142	<0.001
HAM-A *	1.9 (3.1) ^b,c^	13.2 (8.6) ^a,c^	17.8 (8.8) ^a,b^	51.51	2/143	<0.001
Total psychopathology (z)	−2.978 (0.567) ^b,c^	0.277 (1.678) ^a,c^	1.969 (2.046) ^a,b^	106.83	2/143	<0.001
Psychosis	4.0 (0.0) ^b,c^	6.1 (3.5) ^a,c^	7.7 (4.8) ^a,b^	KWT		<0.001
MMSE	28.3 (2.4) ^b,c^	24.5 (4.2) ^a^	24.9 (4.2) ^a^	13.08	2/143	<0.001
Q192R Paraoxonase (PON)1	1/17/22	5/28/23	5/23/24	3.12	4	0.537
CMPAase	42.2 (11.8) ^b,c^	28.4 (5.3) ^a,c^	24.6 (6.9) ^a,b^	58.24	2/145	<0.001
-SH groups	315.5 (59.0) ^a,b^	261.3 (61.7) ^a,c^	232.6 (52.1) ^a,b^	23.62	2/145	<0.001
TRAP	968.7 (143.0) ^b,c^	872.7 (143.8) ^a,c^	772.6 (121.1) ^a,b^	21.87	2/143	<0.001
LOOH	1127.6 (276.9) ^b,c^	1313.0 (340.3) ^a^	1306.0 (325.1) ^a^	5.56	2/143	0.005
AOPP	228.7 (190.5) ^b,c^	356.8 (186.8) ^a^	372.7 (217.0) ^a^	15.45	2/143	<0.001
MDA	2.23 (0.47) ^b,c^	5.17 (1.13) ^a,c^	5.61 (1.17) ^a,b^	129.70	2/143	<0.001
NOx	7.52 (5.94)	6.31 (5.31)	5.97 (6.09)	0.98	2/143	0.367
OSTOX/3ANTIOX	−1.374 (0.572) ^b,c^	0.326 (0.543) ^a,c^	0.740 (0.550) ^a,b^	169.88	2/143	<0.001

All values are shown as mean (SD); ^a,b,c^: Pairwise differences among the subgroups. * Processed in Ln transformation. TLE phenome: a phenotype score extracted from frequency of seizures, controllability of seizures, history of aura, postictal confusion, and temporal lobe epilepsy; TLE-PP phenome: same but with severity of depression, anxiety, and general psychopathology; BMI: body mass index, TUD: tobacco use disorder, HDRS and HAM-A: Hamilton Depression and Anxiety Rating Scales; BPRS: Brief Psychiatric Rating Scale; MMSE: Mini-mental State Examination; CMPA: 4-(chloromethyl)phenyl acetate; TRAP: total radical trapping parameter; LOOH: lipid hydroperoxides; AOPP: advanced oxidation protein products; MDA: malondialdehyde; NOx: nitric oxide metabolites; OSTOX/ANTIOX: index of oxidative stress toxicity/index of antioxidant capacity.

## Data Availability

All of the data is contained within the article.

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
