# Peer review of "A Novel Pathway Phenotype of Temporal Lobe Epilepsy and Comorbid Psychiatric Disorders: Results of Precision Nomothetic Medicine"

_antioxidants, 2022, doi:10.3390/antiox11050803_

Round 1
Reviewer 1 Report
This manuscript by the group of highly reputable researchers is very well designed and written. The conclusions are justified by the results obtained. It is very rare for me, but in this case, I am pleased to say that I have no criticism whatsoever
Author Response
Response: thank you
Reviewer 2 Report
In this manuscript, the authors developed a novel phenotype that integrates the severity of temporal lobe epilepsy (TLE) with psychiatric comorbidities. They also correlated them with oxidative and antioxidant biomarkers. I think that this manuscript is of potential interest for both clinical and non-clinical audience. I really appreciate the fact that the authors are aware of the limitations of the study that discussed in their manuscript. Moreover, the methodologies are appropriate and the results are clearly reported. I have only some points to be addressed:
- Please specify how the PON1 Q192R variant influences the enzyme activity
- Which was the technique for PON1 genotyping? Please add ii in the Method section.
- I should suggest the authors to better discuss the implications of their work in the Conclusion section with a special attention on prognostic and therapeutical implications
Minor:
- Please provide the emails of each author
Author Response
In this manuscript, the authors developed a novel phenotype that integrates the severity of temporal lobe epilepsy (TLE) with psychiatric comorbidities. They also correlated them with oxidative and antioxidant biomarkers. I think that this manuscript is of potential interest for both clinical and non-clinical audience. I really appreciate the fact that the authors are aware of the limitations of the study that discussed in their manuscript. Moreover, the methodologies are appropriate and the results are clearly reported. I have only some points to be addressed:
Point 1: Please specify how the PON1 Q192R variant influences the enzyme activity
Response 1: This is now addressed in the text as:
The PON1 Q192R polymorphism influences the activity of the PON1 enzymes thereby altering their ability to prevent lipid oxidation [20,40], but the direction of this change is substrate dependent [41,42]. RR homozygotes show a greater efficacy detoxifying substrates including paraoxon, CMPA and 5-thiobutil butyrolactone (TBBL) [39,43,44] even though the influence on TBBL (i.e., lactonase activity) is lower (30-50% higher in RR) than paraoxon (100-200% higher in RR) [43]. The Q allozyme is more efficient to detoxify substrates such as diazoxon.
Point 2: Which was the technique for PON1 genotyping? Please add ii in the Method section.
Response 2: addressed in the tekst as:
Because the PON1 polymorphism causes variations in hydrolysis capability, it is possible to stratify genotypes after phenotypic measurement of enzyme activity. We employed 4-(chloromethyl)phenyl acetate (CMPA) (CMPA, Sigma, USA), which is an alternative to the use of the toxic paraoxon, and phenylacetate (Sigma, USA), under high salt conditions, to stratify the functional genotypes of the PON1Q192R polymorphism, namely Q/Q, Q/R, and R/R) [14,20].
Point 2: I should suggest the authors to better discuss the implications of their work in the Conclusion section with a special attention on prognostic and therapeutical implications
Response 2: addressed in the tekst as:
All in all, the severity of the new pathway phenotype of TLE and allocation to the new endophenotype class PP-OS-AO, characterized by increased psychopathology and oxidative damage and lowered antioxidant defenses, signal greater severity of TLE, and increased frequency and uncontrollability of seizures. In animal models, there is some evidence that administration of antioxidants including acetyl-carnitine, L-carnitine, Ginkgo biloba leaf extract, Lantana camara and vanillic acid, exert neuroprotective, anticonvulsant and/or anti-kindling effects [75-79]. There is also some evidence that in TLE patients, treatment with antioxidants such as selenium and vitamin E and C may have some efficacy [80]. Nevertheless, in preclinical models, natural polyphenols have mixed affects and N-acetylcysteine may aggravate seizures and improve depressive-like behaviors [81,82]. The results of the current precision medicine study show that PON1 activity and aldehyde formation may be the most appropriate drug targets to treat severe TLE. Future research should develop and trial new drugs targeting the PON1-aldehyde formation pathway.
Point 3: Please provide the emails of each author
Response 3: mail addresses are provided on page 1 of the MS.